# Graph-informed Neural Point Processes with Monotonic Nets

## Abstract

Multi-class event data is ubiquitous in real-world applications. The recent neural temporal point processes (Omi et al., 2019) have used monotonic nets to model the cumulative conditional intensity to avoid an intractable integration in the likelihood. While successful, they are restricted to single-type events and can easily sink to poor learning results. To address these limitations and to exploit valuable structural information within event participants, we develop a Graph-Informed Neural Point Process (GINPP) that can freely handle multiple event types, greatly improve learning efficiency, and effectively integrate the graph information to facilitate training. First, we find the bottleneck of the previous model arises from the standard softplus transformation over the output of the monotonic net, which enlarges the prediction variations of the monotonic net and increases the training challenge. We propose a shift-scale variant that can significantly reduce the variation and promote the learning efficiency. Second, we use a conditional mark distribution to model multiple event types, without the need for explicitly estimating the intensity for each type. The latter can be much more challenging. Third, we use random walk to collect the neighborhood of each event participant, and use an attention mechanism to update the hidden state of each participant according to the observed events of both the participant itself and its neighborhood. In this way, we can effectively leverage the graph knowledge, and scale up to large graphs. We have shown the advantage of our approach in both ablation studies and real-world applications.

## 1 Introduction

Real-world applications often involve multi-class events. For example, 911 calls seek for a variety of helps, traffic records include different types of accidents, and among social network users are various types of interactions (tweeting, following, poking, *etc*.). Neural temporal point processes (*e.g.*, (Du et al., 2016; Mei and Eisner, 2017; Zhang et al., 2020a; Zuo et al., 2020)) are a family of powerful methods for event modeling and prediction, which use neural networks (NN) to model the intensity of events and can flexibly estimate the complex dependencies among the observed events. However, due to the use of NNs, the cumulative (*i.e.*, integral of) conditional intensity in the point process likelihood is often analytically intractable, and demand a complex, expensive approximation. To bypass this issue, the recent work Omi et al. (2019) uses a monotonic net (Sill, 1997; Chilinski and Silva, 2020) to model the monotonically increasing cumulative intensity to avoid the integration, and the intensity is obtained by simply taking the derivative. To ensure the positiveness, a softplus transformation is applied to the output of the monotonic net. Despite the elegance and success, this method only supports single-type events. More important, it often suffers from inefficient learning and easily falls into poor performance.

In this paper, we propose GINPP, a graph-informed neural point process model to overcome these problems, and to further utilize the valuable structural knowledge within the event participants, which is often available in practice. The major contributions of our work are listed as follows.

- First, we investigate the learning challenge of (Omi et al., 2019), and find the bottleneck arises from the softplus transformation over the monotonic net prediction to ensure positiveness. To obtain an output slightly above zero, the standard softplus demands the input, *i.e.*, the monotonic net prediction, must be negative and have much greater scales. Hence, a small output range can cause a much wider input (monotonic net prediction) range, which

are biased toward the negative domain. The large variation of the prediction scale makes the estimation of the monotonic net much more difficult and inefficient.

- Second, we propose a shift-scale variant of the softplus function, where the scale controls the shape and the shift controls the position. By setting these two hyperparameters properly, the required input range can be greatly shrunk, and get close to the output range. Accordingly, the variations of the prediction scales is significantly reduced, and the learning of the monotonic net is much easier and more efficient.
- Third, we construct a marked point process for multi-class events. By introducing a conditional mark distribution, we can freely handle different event types and only need a single-output monotonic net, which models the unified cumulative conditional intensity. This is more efficient and convenient than a naive extension that separately estimates the cumulative intensity for each particular event type.
- Fourth, to incorporate the graph structure in training, we use random walk to collect the neighborhood for each participant. We use an attention mechanism to update the hidden state of each participant, according to the observed events of not only the participant itself, but also its neighborhood. In this way, the estimation of hidden state can be improved with enriched observations, and the event dependencies can be more comprehensively captured. The random walk further enables us to scale to large graphs. Accordingly, we develop an efficient, scalable stochastic mini-batch learning algorithm.

For evaluation, we first examined GINPP in ablation studies. We tested the performance of the monotonic net with our shift-scale softplus transformation in learning two benchmark functions: one is monotonic and the other is not. Our method converges fast, accurately learns the first function, and finds a close monotonic approximation to the second function. By contrast, with the standard softplus, the learning is saturated early at large loss values and the estimation is much worse. Then, we tested on a synthetic bi-type event dataset. GINPP accurately recovered the intensity for each event type via the learned overall intensity and the mark distribution. Next, we evaluated GINPP in six real-world benchmark datasets. We examined the accuracy in predicting the time and type of future events. In both tasks, GINPP consistently outperforms all the competing methods. Even without incorporating the graphs, GINPP still achieves better accuracy. When the graph structure is available, GINPP improves the accuracy further.

## 2 Background

**Temporal Point Process (TPP)** is a general mathematical framework for event modeling (Daley and Vere-Jones, 2007). A TPP is specified by the conditional intensity (or rate) of the events. Suppose we have $K$ types of events, and denote by $\lambda_k(t)$ the conditional intensity for event type $k$. Given a sequence of observed events and their types, $\Gamma = [(t_1, s_1), \ldots, (t_N, s_N)]$, where $t_n$ is the timestamp and $s_n$ is the type of each event $n$ ($1 \le s_n \le K$, $t_n \le t_{n+1}$). The likelihood of the TPP is given by

$$p(\Gamma) = \prod_{k=1}^{K} \exp\left(-\int_0^T \lambda_k(t)\mathrm{d}t\right) \cdot \prod_{n=1}^{N} \lambda_{s_n}(t_n), \tag{1}$$

where $T$ is the entire span of the observed events. One popular TPP is the homogeneous Poisson process, which assumes each conditional intensity $\lambda_k(t)$ is a time-invariant constant $\lambda_k^0$, and has nothing to do with previous events $\{(t_n, s_n) | t_n < t\}$. While simple and convenient, Poisson processes ignore the complex relationships among the events. The Hawkes process (Hawkes, 1971) is more expressive in that it models the excitation effect among the events,

$$\lambda_k(t) = \lambda_k^0 + \sum_{t_n < t} \rho_{s_n \to k}(t - t_n) \tag{2}$$

where $\lambda_k^0 \ge 0$ is the background rate, $\rho_{s_n \to k}(\Delta) > 0$ is the triggering kernel, and quantifies how much contribution the past event at $t_n$, of type $s_n$, makes to trigger a new event of type $k$ to occur at $t$. The most commonly used triggering kernel is an exponential kernel, which assumes an exponential decay of the excitation effect along with the time lag $\Delta$.

**Neural Temporal Point Process.** Hawkes processes only account for additive, excitation effects, and are inadequate to capture various complex event dependencies. To overcome this limitation, recent works (Du et al., 2016; Mei and Eisner, 2017) use neural networks to model the conditional intensity. Typically, a recurrent neural network (RNN) is used to capture the complex event dependencies. For

each event $n$, we introduce a hidden state $\mathbf{h}_n$, which is computed according to the previous state $\mathbf{h}_{n-1}$, the current time, event type, and other input features. An illustrative example is

$$\mathbf{h}_n = \text{RNN-Cell}(\mathbf{h}_{n-1}, t_n, s_n). \tag{3}$$

Then we obtain the conditional intensity through a positive transformation over the hidden state, *e.g.*, $\lambda_k(t) = f(\mathbf{w}_k^\top \mathbf{h}_n)$. We then substitute the intensity into (1), and maximize the likelihood to estimate the model parameters.

Although the NN modeling of $\lambda_k(t)$ greatly increases the model capacity/expressivity, it makes the cumulative intensity in the likelihood (1), namely $\int_0^T \lambda_k(t) \mathrm{d}t$, analytically intractable to compute. We have to use approximations, such as Monte-Carlo sampling and numerical quadrature, which can be expensive and complex. To sidestep this issue, Omi et al. (2019) instead modeled the cumulative conditional intensity with the RNN output,

$$\phi(\mathbf{h}_{n-1}, t) = f_{sp} \left( \text{MNet} \left( \mathbf{h}_{n-1}, t \right) \right) = \int_{t_{n-1}}^t \lambda(\tau) \mathrm{d}\tau, \tag{4}$$

where $t_{n-1} \leq t \leq t_n$, $\mathbf{h}_{n-1}$ is the RNN state corresponding to the last observed event, MNet is a monotonic net (Sill, 1997; Chilinski and Silva, 2020), which guarantees the output is monotonically increasing along with the input time $t$ and hence is consistent with the cumulative intensity, and $f_{sp}$ is the softplus function, $f_{sp}(\cdot) = \log(1 + \exp(\cdot))$, which is to ensure the positiveness. Note $f_{sp}$ is also monotonically increasing, and the transformation with $f_{sp}$ does not change the monotonicity on $t$. Since Omi et al. (2019) only considered single-type events, we omit the subscript $k$ and denote the single conditional intensity by $\lambda(t)$. Given (4), we can obtain the conditional intensity by taking the derivative, $\lambda(t_n) = \left. \frac{\partial \phi(\mathbf{h}_{n-1}, t)}{\partial t} \right|_{t=t_n}$. The likelihood of an event sequence $[t_1, \ldots, t_N]$ is

$$p(t_1, \ldots, t_N) = \prod_{n=1}^N \exp \left( -\int_{t_{n-1}}^{t_n} \lambda(t) \mathrm{d}t \right) \cdot \exp \left( -\int_{t_N}^T \lambda(t) \mathrm{d}t \right) \cdot \prod_{n=1}^N \lambda(t_n)$$

$$= \prod_{n=1}^N \phi(\mathbf{h}_{n-1}, t_n) \cdot \phi(\mathbf{h}_N, T) \cdot \prod_{n=1}^N \left. \frac{\partial \phi(\mathbf{h}_{n-1}, t)}{\partial t} \right|_{t=t_n} \tag{5}$$

where $t_0 = 0$ and $\mathbf{h}_0$ is initial state of the RNN. Since there is no integration, the computation and optimization is much easier and more convenient, especially with automatic differentiation libraries.

## 3 Model

Although the model of (Omi et al., 2019) is smart and successful, it only supports single-type events. More important, we found that it often suffers from inefficient learning and easily falls into poor performance. To address these issues and to further take advantage of the structural knowledge within the event participants, we develop GINPP, a graph-informed neural point process model based on the monotonic net, presented as follows.

Specifically, we assume our dataset includes $K$ types of events, which were launched by $M$ participants. Each observed event sequence is a series of mixed-type events launched by a particular participant. For example, in online social media, a tweeter account can be viewed as a participant, which can launch a series of events of different types: tweeting, retweeting, replying, like, direct messaging, *etc*. We denote the event sequence of participant $m$ by $\Gamma^m = [(t_1^m, s_1^m), \ldots, (t_{N_m}^m, s_{N_m}^m)]$ where each event type $s_n^m \in \{1, \ldots, K\}$ ($1 \leq n \leq N_m$). Among the $M$ participants is a graph structure that encodes their correlations, denoted by $\mathcal{G} = (\mathbb{E}, \mathbb{V})$ where $\mathbb{V} = \{1, \ldots, M\}$ is the vertex set, and $\mathbb{E} = \{(i, j)\}$ is the edge set.

First, we consider extending (Omi et al., 2019) to support multiple event types. A straightforward extension is to follow the idea of (Mei and Eisner, 2017) and expand the output dimension of the monotonic net to $K$ (see (4)) for each participant (vertex) $m$. We then apply an element-wise softplus transformation to obtain the cumulative conditional intensity for each event type $k$, namely, $\phi_k(\mathbf{h}_{n-1}^m, t) = f_{sp} \left( \text{MNet} \left( \mathbf{h}_{n-1}^m, t \right) [k] \right)$, where $\mathbf{h}_{n-1}^m$ is the hidden state of the vertex $m$. However, this method will increase the learning challenge of the monotonic net, because $K$ monotonic constraints have to be satisfied simultaneously. To circumvent this issue, we construct a marked point

process (Daley and Vere-Jones, 2007), where the event type is considered as a mark of the event. As in (4), we still use a single-output monotonic net, but to model a global cumulative conditional intensity $\phi(\mathbf{h}_{n-1}^m, t)$. We then introduce a mark distribution to sample the event type according to the last state $\mathbf{h}_{n-1}^m$ and the time lag $\Delta_t = t - t_{n-1}^m$,

$$p(s = k|t) \propto \exp\left(\mathbf{u}_k^\top \boldsymbol{\beta}(\mathbf{h}_{n-1}^m, \Delta_t)\right) \tag{6}$$

where $\boldsymbol{\beta}(\cdot)$ is the output of a neural network, and $\mathbf{u}_k$ is the embedding of event type $k$, which will be jointly estimated during training. The conditional intensity is therefore given by $\lambda_k^m(t) = \lambda^m(t)p(s = k|t)$ where $\lambda^m(t) = \frac{\partial \phi(\mathbf{h}_{n-1}^m, t)}{\partial t}$ is the global conditional intensity. The likelihood of the event sequence $\Gamma^m = [(t_1^m, s_1^m), \ldots, (t_{N_m}^m, s_{N_m}^m)]$ is therefore a minor adjustment of (5),

$$p(\Gamma^m) = \exp\left(-\int_0^T \left[\sum_{k=1}^K \lambda_k^m(t)\right] dt\right) \prod_{n=1}^{N_m} \lambda_{s_n^m}^m(t_n^m)$$

$$= \prod_{n=1}^{N_m} \phi(\mathbf{h}_{n-1}^m, t_n^m) \cdot \phi(\mathbf{h}_{N_m}^m, T) \cdot \prod_{n=1}^{N_m} \left( p(s = s_n^m | t_n^m) \left. \frac{\partial \phi(\mathbf{h}_{n-1}^m, t)}{\partial t} \right|_{t = t_n^m} \right). \tag{7}$$

This simple modification enables us to freely model multiple types of events, but sidesteps the difficulty of learning a multi-output monotonic net. Empirically, we found our method is much more effective.

Second, we investigate the learning challenge of (Omi et al., 2019). We find the bottleneck arises from the standard softplus transformation $f_{sp}$ in (4), which enlarges the prediction scales of the monotonic net and increases the training difficulty. Specifically, as a continuous relaxation of the ReLU activation, the output $f_{sp}(x)$ is approximately equal to the input $x$ only when the output is relatively big, $e.g.$, $3.05 = f_{sp}(3)$. By contrast, when the output is small, $e.g.$, close to zero, the input $x$ is a negative number with a much larger scale, $e.g.$, $0.0009 = f_{sp}(-7)$. This can be seen from the inverse, $f_{sp}^{-1}(y) = \log(e^y - 1)$. As shown in Fig. 1a, when $y$ is close to zero, $e^y - 1$ is close to zero, and the inverse function varies violently. As a consequence, a small output range requires a much wider input range, and the monotonic net's prediction has to cover this input range. For example, an output range $[10^{-4}, 3]$ corresponds to the input range $[-9.21, 2.95]$. Hence, the variation of the monotonic net's prediction is greatly enlarged. This can be further verified from a probabilistic analysis. Suppose the training output follows a uniform distribution in $[0, 3]$. This is reasonable, because in practice, we often normalize the data to avoid their scales being too large for better numerical stability and optimization efficiency. We then look into the corresponding input distribution of the standard softplus. As shown in Fig. 1b (the blue

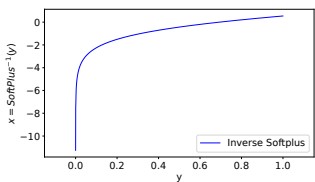

(a) Inverse softplus

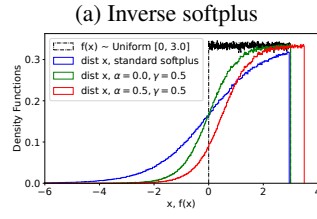

(b) Density curve

Figure 1: Inverse of the standard softplus and the input distribution for shift-scale softplus.

line), the input distribution is much wider, and includes a long, fat tail over the negative domain. That means, the prediction of the monotonic net has to fit many negative values with much larger scales, which can bring back the troubles in optimization efficiency and numerical stability. To alleviate this issue, we propose a shift-scale variant of the softplus function,

$$\hat{f}_{sp}(x; \alpha, \gamma) = \gamma \log(1 + \exp(\frac{x - \alpha}{\gamma})) \tag{8}$$

where $\alpha$ and $\gamma$ are two hyperparameters, shift and scale. When $\alpha > 0$, the function body is moved toward the right of the x-axis and hence the negative input range needed to obtain close-to-zero outputs is shrunk. Furthermore, when we choose $\gamma \in (0, 1)$, the absolute value of the input is amplified. That means, to achieve the same output, $e.g.$, 0.0009, the scale of the input $x$ — $i.e.$, the output of the monotonic net — can be greatly decreased. Therefore, both $\alpha$ and $\gamma$ can shrink the input range, and reduce the fat tail of the distribution over the negative inputs, so as to make monotonic net learning easier and more efficient. Fig. 1b shows the input distribution with $\alpha = 0, \gamma = 0.5$ and $\alpha = \gamma = 0.5$. In both cases, the distribution over the large negative inputs is greatly reduced. The reduction with $\alpha = \gamma = 0.5$ is more significant. Our experiments have verified the improvement of learning with our shift-scale softplus function (see Sec. 6.1).

Finally, to incorporate the valuable graph knowledge, we use an attention mechanism to model the RNN states at each vertex (participant), based on the observed events occurred on both the vertex itself and its neighborhood (see Fig. 2). Specifically, we introduce an embedding vector $\mathbf{v}_m$ to represent each vertex $m$. To compute the RNN states at each vertex $m$, we flatten the observed event sequences on $m$

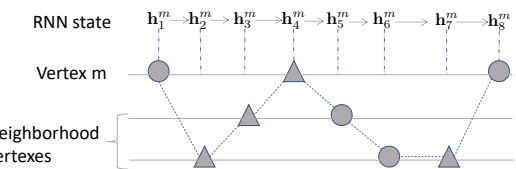

RNN state

Vertex m

Neighborhood
vertexes

Figure 2: State update for a vertex.

and its neighborhood $\mathcal{B}_m$ to obtain a single sequence $\widehat{\Gamma}^m = [(t_1, s_1, r_1), \ldots, (t_N, s_N, r_N)]$, where $t_1 \leq \ldots \leq t_N$, $\{s_n\}_{n=1}^N$ are the event types, $\{r_n\}_{n=1}^N$ the vertex indices, and each $r_n \in \{m\} \cup \mathcal{B}_m$. For vertex $m$, we introduce a hidden state $\mathbf{h}_n^m$ at each event $(t_n, s_n, r_n)$. We update $\mathbf{h}_n^m$ by

$$\mathbf{g}_n^m = \text{RNN-Cell}(\mathbf{h}_{n-1}^m, \mathbf{u}_{s_n}, t_n - t_{n-1}), \tag{9}$$

$$a_n = \sigma(\tau \cdot \mathbf{v}_m^\top \mathbf{v}_{r_n}), \tag{10}$$

$$\mathbf{h}_n^m = a_n \cdot \mathbf{g}_n^m + (1 - a_n) \cdot \mathbf{h}_{n-1}^m \tag{11}$$

where $\sigma$ is the sigmoid function, $a_n$ is an attention score computed from the inner product of the vertex embeddings, and $\tau > 0$ adjusts how much the score is leaning toward 0 or 1 (in our experiment we set $\tau = 2.0$). We can see that even if the event did not occur on vertex $m$, *i.e.*, $r_n \neq m$, we can still update the state on $m$ according to its similarity to the neighborhood vertex $r_n$, *i.e.*, the attention score $a_n$. This score reflects how much the vertex $m$ should pay attention to the event happened on the neighborhood to update itself. If vertex $m$ and $r_n$ are very dissimilar, *i.e.*, $a_n \approx 0$, then there is almost no update. In this way, the correlation between the vertexes (participants), which are reflected by the graph structure, are integrated into our model to bolster the estimation of the cumulative conditional intensity (see (4)), better fit the observed data and improve the prediction. When there is no graph structure, we have all $r_n = m$ and $a_n \approx 1$, and we return to the standard RNN updates.

## 4 Algorithm

Given a collection of observed event sequences on all the vertexes (participants), $\mathcal{D} = \{\mathcal{S}_1, \ldots, \mathcal{S}_L\}$, where each $\mathcal{S}_l = \{\Gamma_l^1, \ldots, \Gamma_l^M\}$ $(1 \leq l \leq L)$ and each $\Gamma_l^m$ is an event sequence on vertex $m$ $(1 \leq m \leq M)$, we develop a stochastic mini-batch learning algorithm to estimate the vertex embeddings $\{\mathbf{v}_j\}_{j=1}^M$, the event type embeddings $\{\mathbf{u}_k\}_{k=1}^K$, the parameters of the RNN and monotoic net (see (9) and(4)), and the NN parameters for mark distribution (see (6)).

Specifically, to be scalable to a large graph $\mathcal{G}$, at each step, we use random walk to sample a small set of vertexes $\mathcal{B}$. We view the vertexes in $\mathcal{B}$ are neighborhood to each other. Thereby, the neighborhood includes not only the vertexes that are directly connected, but also the ones connected via a short path. In this way, we can exploit more abundant local structural information. Next, to handle a large number sequences $L$, we randomly select one sequence set $\mathcal{S}_l$ $(l \in \{1, \ldots, L\})$, and use the subset of sequences on the sampled vertexes, $\{\Gamma_l^m | m \in \mathcal{B}\}$, to compute the likelihood (see (7)) and its gradient. We use this gradient as a stochastic gradient to update the model parameters. We can apply any stochastic optimization algorithm. Our stochastic training is summarized in Algorithm 1.

---

**Algorithm 1** GINPP $(\mathcal{D}, E, \mathcal{G}, \alpha, \gamma)$

---

   Initialize the model parameters
   **repeat**
      Randomly shuffle the sequence sets in $\mathcal{D} = \{\mathcal{S}_1, \ldots, \mathcal{S}_L\}$
      **for** each set $\mathcal{S}_l$ **do**
         Random walk over the graph $\mathcal{G}$ to sample a small set of vertexes $\mathcal{B}$.
         Pick the subset of sequences associated with $\mathcal{B}$: $\mathcal{A} = \{\Gamma_l^m | m \in \mathcal{B}\}$
         Compute the gradient of the likelihood on $\mathcal{A}$ according to (7).
         Update the model parameters with the gradient (*e.g.*, ADAM)
      **end for**
   **until** $E$ epochs are done

---

## 5 Related Work

Poisson process is a popular tool for event data analysis, *e.g.*, (Lloyd et al., 2015; Schein et al., 2015; 2016; 2019), but its independent increment assumption ignores the event dependencies or interactions. Many works therefore propose to use Hawkes processes (HPs) (Hawkes, 1971) to capture the mutual excitation effects among the events, such as (Blundell et al., 2012; Tan et al., 2016; Linderman and Adams, 2014; Du et al., 2015; He et al., 2015; Wang et al., 2017; Yang et al., 2017a; Xu and Zha, 2017; Xu et al., 2018). A series of works improve the learning with HPs, such as nonparametric kernel estimation (Zhou et al., 2013; Zhang et al., 2020b; Zhou et al., 2020), short doubly-censored event sequences (Xu et al., 2017), Granger causality (Xu et al., 2016) and online estimation (Yang et al., 2017b). Another recent line of works (Zhe and Du, 2018; Pan et al., 2020; Wang et al., 2020) uses the HP framework for high-order structure decomposition and representation learning.

In order to estimate more complex event dependencies, recent research has attempted to use neural networks to construct temporal point processes. Du et al. (2016) used an RNN to construct a marked temporal point process, where the conditional intensity is formulated as a linear transformation of the RNN state and the time difference then through an exponential transformation to ensure the positiveness. Mei and Eisner (2017) proposed a continuous-time LSTM (Hochreiter and Schmidhuber, 1997) to model the conditional intensity and used the softplus transformation to ensure the positiveness. To handle the intractable integration in the point process likelihood, Mei and Eisner (2017) used a Monte-Carlo approximation. The recent works (Zhang et al., 2020a; Zuo et al., 2020) use the attention mechanism (Vaswani et al., 2017; Bahdanau et al., 2014) to replace the RNN framework, but they still have to approximate the integration in the likelihood. Omi et al. (2019) bypassed this problem by feeding the RNN states into a monotonic net (Sill, 1997; Chilinski and Silva, 2020) to directly model the cumulative conditional intensity, so we do not need to explicitly compute the integration. The monotonic net is typically a multi-layer perceptron but imposes the nonnegative constraint over the weights during learning to fulfill the monotonicity. While successful, the model of Omi et al. (2019) only supports single-type events, which can be limited in practice. Our work GINPP extends their model to support multi-type events with the marked point process framework (Daley and Vere-Jones, 2007), which is simple and efficient. Note that Du et al. (2016) also used the marked point process framework. GINPP uses the RNN framework to model the hidden state due to the stable and excellent performance, but an attention mechanism is used to incorporate the graph structure into the state computation. Hence, GINPP can be viewed as a hybrid approach. It is straightforward to extend GINPP to a full attention model.

Other works include Zhou et al. (2021) that extends HPs with a mixture of shifted Beta densities, ODE based models (Rubanova et al., 2019), neural network influence kernels (Zhu et al., 2021a) , deep Fourier kernels (Zhu et al., 2021b), the intensity-free point process learning (Shchur et al., 2019) that models the time difference between successive events, *etc*. Based on the intensity-free framework (Shchur et al., 2019), Zhang et al. (2021) also proposed a neural point process model with a prior graph incorporated. But their goal is to infer the labels of nodes in the graph to detect clusters and anomalies, rather than predict the occurrence and type of new events. The recent work of Pan et al. (2021) develops a nonparametric decaying model of the temporal inference, and can explicitly recover various excitation and inhibition effects, and their decay patterns among the events. While flexible and interpretable, it still needs to approximate the cumulative conditional intensity, which is done via Gauss-Laguerre quadrature.

## 6 Experiment

### 6.1 Ablation Study

We first performed an ablation study to confirm the effectiveness of our shift-scale softplus transformation (8). To this end, we tested with two functions:

$$g_1(x) = 0.5 \cdot \sigma\left(5(x-1)\right), \quad g_2(x) = 0.3 \cdot \left(0.5\sin(10x)e^{-x/2} + e^x - 1\right) - 0.2 \quad (12)$$

where $g_1(x) > 0$ monotonically increases with the input $x$ and $g_2$ does not. We used a monotonic net plus the shift-scale softplus transformation to learn the two functions. The monotonic net includes two layers, with 256 neurons per layer, and tanh activation. We uniformly sampled 50 training points from $x \in [0, 2]$. We implemented the model with TensorFlow, and used Adam for stochastic optimization. The learning rate was set to $10^{-3}$. We set $\alpha = 0, \gamma = 0.2$ and $\alpha = 0.5, \gamma = 0.2$. We also tested with the standard softplus We show the learning curves and estimated the functions in

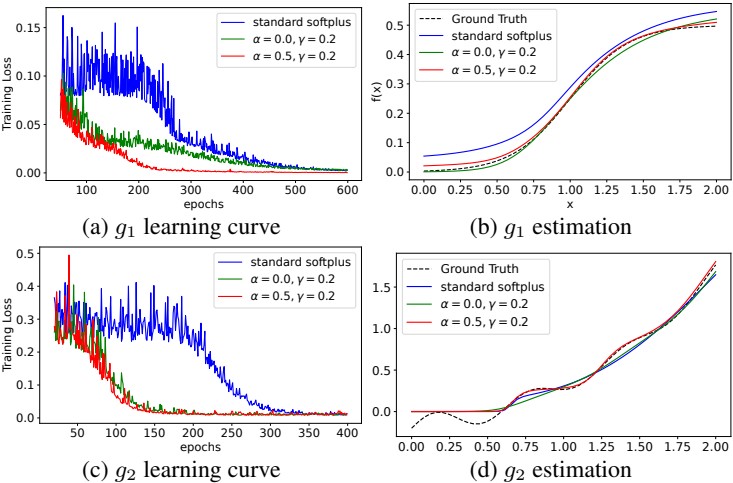

(a) $g_1$ learning curve

(b) $g_1$ estimation

(c) $g_2$ learning curve

(d) $g_2$ estimation

Figure 3: The learning curves and estimation results of monotonic net with softplus transformation.

Fig. 3. As we can see, with our shift-scale softplus, the learning converges much faster (see Fig. 3a and Fig. 3c). With the standard softplus, the learning fell into a relative big loss first, and stayed there for quite a long time before the new improvement started to happen. By contrast, during the stage that the standard softplus was stuck, our shift-scale softplus had driven the training loss to drop fast and converged (see the red line). This shows that the shift-scale version can greatly boost the learning efficiency. Note that for standard softplus, the long stucking stage can signal a wrong message that the training can stop. The estimated functions with our shift-scale softplus are also more accurate. From Fig. 3b and 3d, our learned functions are quite close to the ground-truth. In Fig. 3b, our estimation ($\alpha = 0.5, \gamma = 0.2$) almost overlaps with the ground-truth. By contrast, the estimation with the standard softplus (the blue line) has a clear deviation. Therefore, it shows that our shift-scale softplus not only bolster the learning efficiency but also leads to better learning results.

Next, we examined if GINPP can recover the ground-truth conditional intensity. To this end, we generated a synthetic dataset of bi-type events, where type $0$ events excite type $1$ events while type $1$ inhibits type $0$, and events of the same type do not influence on each other,

$$\rho_{0 \to 1}(\Delta) = \max(1.0 - 0.05\Delta^2, 0),$$
$$\rho_{1 \to 0}(\Delta) = -0.5 \exp(-0.5\Delta).$$

We then substitute $\rho_{0 \to 1}$ and $\rho_{1 \to 0}$ into (2) (background rate is zero). To ensure the positiveness, we applied a softplus transformation to obtain the conditional intensity. We used Thinning algorithm Lewis and Shedler (1979) to sample 10K sequences for training and 1K for validation. Each sequence consists of $64$ events. We evalu-

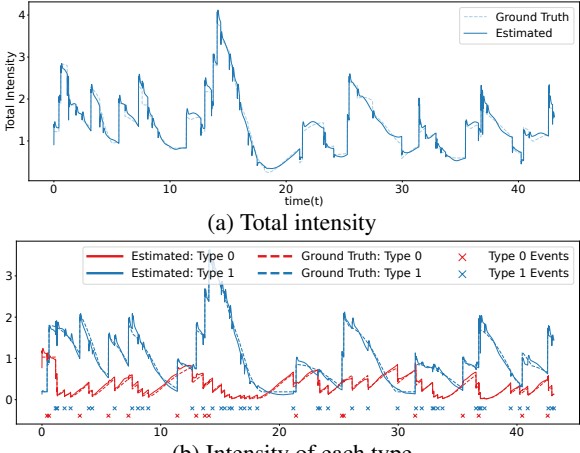

(a) Total intensity

(b) Intensity of each type

Figure 4: Estimation of the conditional intensity.

ated the likelihood of the validation set after each training epoch, and halted the training when no improvement was observed (*i.e.*, early stopping). We used the GRU cell for our RNN component. The dimension of the hidden state and event type embeddings was set to 64 and 16, respectively. For the monotonic net component, we used two hidden layers, 256 neurons per layer, and tanh activation. For the NN in the mark distribution (see (6)), we used one hidden layer, with 256 neurons and RELU activation function. We set $\alpha = \gamma = 0.5$ for the shift-scale softplus. We used Adam for stochastic optimization with learning rate $10^{-3}$ and mini-batch size $64$. The maximum number of epochs was set to 100. Fig. 4a shows the estimated total conditional intensity, and we can see it matches the ground-truth quite closely. Then Fig. 4b reports the derived conditional intensity for type $0$ and type $1$ from the learned mark distribution. As we can see, GINPP can also accurately recover the intensity for each event type, although it does not explicitly estimate them. The results have confirmed the capability of our multi-type event model.

## 6.2 Predictive Performance

Next, we evaluated the performance of GINPP in predicting the time and type of future events. To this end, we used six real-world benchmark datasets, where the first four do not include graph information and the remaining two are associated with graphs. (1) Retweet (Zhao et al., 2015), including 24K event sequences and three event types: retweeting by "small" , "medium" and "large" retweeters. (2) SO (Du et al., 2016), $6,633$ awarding sequences in the Q/A site Stack Overflow. The event type is the award, such as "Nice Question", "Guru", and "Great answer". There are 22 event types. (3) MIMIC (Du et al., 2016), clinical visit events from 650 anonymous patients in seven years, and the event type is the diagnosis outcome. There are 75 outcomes. (4) Social[1], 48.9K student activity events in the campus of a university. The event type corresponds to the campus location of the event and we have 50 event types, and 1,614 event sequences happened among 51 students. (5) 911Call[2], 251K emergence calls in Montgomery County of Pennsylvania from year 2015 to 2017. There are three types of calls (events): EMS, FIRE and Traffic. We have collected 6,187 event sequences from 73 zones. We viewed each zone as a vertex, and added an edge if two zones are neighboring each other. (6) SLCCrimel[3], 48.7K criminal events happened in 23 zones of Salt Lake City. There are 25 event types (*e.g.*, robbery, murder and arson) and 1,403 sequences. We viewed each zone as a vertex and created the graph according to the geographic neighboring relationship.

We compared with the following popular and/or state-of-the-art approaches: (1) PP, simple homogeneous Poisson process, (2) Hawkes process (HP) with an exponential triggering kernel, (3) Recurrent Marked Temporal Point Process (RMTPP) (Du et al., 2016), (4) Neural Hawkes Processes (NeuralHP) (Mei and Eisner, 2017), (5) Self-Attentive Hawkes Process (SAHP) (Zhang et al., 2020a) that uses the self-attention mechanism to model the conditional intensity for each event type, (6) Transformer Hawkes process (TRHP) (Zuo et al., 2020) that also uses the attention mechanism to model the intensity. In addition, we compared with (7) Simple Statistics (SS) that predicts the occurrence time of new events with the average lag between successive events, and predicts the new event type with the most frequent type of the observed events. We used the original implementation of NeuralHP (`https://github.com/HMEIatJHU/neurawkes`), SAHP (`https://github.com/QiangAIResearcher/sahp_repo`), TRHP (`https://github.com/SimiaoZuo/Transformer-Hawkes-Process`), and a popular open-source implementation of RMTPP (`https://github.com/woshiyyya/ERPP-RMTPP`). For GINPP, we employed the same setting as in the ablation study (see Sec. 6.1). In addition, we set the dimension of the vertex embeddings to 16. To perform random walk, we randomly selected an initial vertex, and each step randomly hopped to a vertex that connects to the current vertex. The probability is the inverse of the degree of the current vertex. We maintained the visited vertex set until the number reached to the mini-batch size. Except SS, all the methods used ADAM for stochastic optimization with learning rate $10^{-3}$, and the mini-batch size was chosen from $\{8, 16, 32\}$. We used the default settings of all the other methods, and early stopping for every method.

We randomly split each dataset into $70\%$ for training, $10\%$ for validation, and $20\%$ for testing. Each method was used to predict the occurrence time and type of the last event in each test sequence. We repeated the experiment for five times, and computed the average root-mean-square-error (RMSE) and classification accuracy (ACC) for time and type predictions, respectively. We computed the standard deviation. For all the datasets, we also ran our method without graphs (*i.e.*, we set an empty $\mathcal{G}$), denoted by GINPP-1. The results are reported in Table 1. As we can see, in every dataset, GINPP outperforms all the competing methods, in many cases by a large margin. It shows our method is superior in both event time and type prediction. When the graph knowledge is available, GINPP is better than our method not incorporating the graph structure, *i.e.*, GINPP-1. The results show that the graph structure can further facilitate the training and prediction, and our attention method is effective.

Finally, we investigated the attention score between the vertexes after training. Specifically, we examined three zones in *911Call* dataset: Zone 2, 52 and 67. Fig. 5a shows the locations of all the zones. The points represent the observed events on those zones. The color represents the region of each zone. Then Fig 5b, 5c and 5d show the attention scores between the all the event locations and the events in zone 2, 52 and 67, respectively. The color indicates the magnitude; see the color bar. We can see that in many cases, when the event locations are in the neighboring zones, the attention

---

[1] `http://realitycommons.media.mit.edu/SocialEvolutionData.html`
[2] `https://www.kaggle.com/datasets/mchirico/montcoalert`
[3] `https://opendata.utah.gov/browse?category=Public%20Safety`

| *RMSE* | Retweet | SO | MIMIC | Social | 911Call | SLCCrime |
|---|---|---|---|---|---|---|
| SS | $33.241 \pm 0.28$ | $1.561 \pm 0.02$ | $1.274 \pm 0.07$ | $11.643 \pm 3.28$ | $21.754 \pm 0.52$ | $8.024 \pm 0.30$ |
| PP | $34.294 \pm 0.33$ | $1.178 \pm 0.02$ | $1.132 \pm 0.05$ | $7.630 \pm 1.08$ | $19.119 \pm 0.24$ | $6.340 \pm 0.36$ |
| HP | $32.557 \pm 0.40$ | $1.240 \pm 0.01$ | $1.002 \pm 0.05$ | $7.670 \pm 1.03$ | $18.380 \pm 0.24$ | $6.382 \pm 0.36$ |
| RMTPP | $47.704 \pm 0.29$ | $1.656 \pm 0.04$ | $1.015 \pm 0.04$ | $7.966 \pm 1.06$ | $22.031 \pm 1.06$ | $8.873 \pm 0.37$ |
| NeuralHP | $34.912 \pm 0.34$ | $1.173 \pm 0.01$ | $1.026 \pm 0.03$ | $7.740 \pm 1.07$ | $20.790 \pm 0.23$ | $7.498 \pm 0.40$ |
| SAHP | $34.894 \pm 0.35$ | $1.565 \pm 0.05$ | $1.035 \pm 0.04$ | $7.941 \pm 1.07$ | $20.398 \pm 0.30$ | $7.970 \pm 0.40$ |
| TRHP | $34.055 \pm 0.36$ | $1.127 \pm 0.02$ | $1.071 \pm 0.03$ | $7.912 \pm 1.03$ | $20.919 \pm 0.26$ | $7.337 \pm 0.48$ |
| GINPP-1 | $\mathbf{32.258 \pm 0.31}$ | $\mathbf{1.112 \pm 0.01}$ | $\mathbf{0.874 \pm 0.03}$ | $\mathbf{7.616 \pm 1.08}$ | $16.155 \pm 0.27$ | $6.072 \pm 0.31$ |
| GINPP | - | - | - | - | $\mathbf{15.446 \pm 0.28}$ | $\mathbf{5.993 \pm 0.29}$ |
| *ACC* | | | | | | |
| SS | $0.549 \pm 0.002$ | $0.366 \pm 0.010$ | $0.305 \pm 0.021$ | $0.122 \pm 0.006$ | $0.526 \pm 0.008$ | $0.222 \pm 0.006$ |
| PP | $0.549 \pm 0.002$ | $0.366 \pm 0.010$ | $0.186 \pm 0.050$ | $0.110 \pm 0.005$ | $0.544 \pm 0.008$ | $0.181 \pm 0.011$ |
| HP | $0.540 \pm 0.014$ | $0.357 \pm 0.007$ | $0.294 \pm 0.041$ | $0.123 \pm 0.009$ | $0.511 \pm 0.013$ | $0.216 \pm 0.013$ |
| RMTPP | $0.575 \pm 0.006$ | $0.376 \pm 0.008$ | $0.848 \pm 0.018$ | $0.509 \pm 0.008$ | $0.531 \pm 0.005$ | $0.203 \pm 0.009$ |
| NeuralHP | $0.574 \pm 0.014$ | $0.383 \pm 0.007$ | $0.718 \pm 0.041$ | $0.556 \pm 0.009$ | $0.511 \pm 0.013$ | $0.218 \pm 0.013$ |
| SAHP | $0.497 \pm 0.027$ | $0.305 \pm 0.025$ | $0.337 \pm 0.036$ | $0.053 \pm 0.015$ | $0.293 \pm 0..021$ | $0.041 \pm 0.016$ |
| TRHP | $0.541 \pm 0.004$ | $0.375 \pm 0.008$ | $0.768 \pm 0.015$ | $0.485 \pm 0.007$ | $0.532 \pm 0.008$ | $0.220 \pm 0.011$ |
| GINPP-1 | $\mathbf{0.607 \pm 0.004}$ | $\mathbf{0.401 \pm 0.009}$ | $\mathbf{0.863 \pm 0.015}$ | $\mathbf{0.616 \pm 0.011}$ | $0.550 \pm 0.005$ | $0.228 \pm 0.011$ |
| GINPP | - | - | - | - | $\mathbf{0.553 \pm 0.005}$ | $\mathbf{0.233 \pm 0.009}$ |

Table 1: Predictive performance of the time (RMSE) and type (ACC) of the future events.

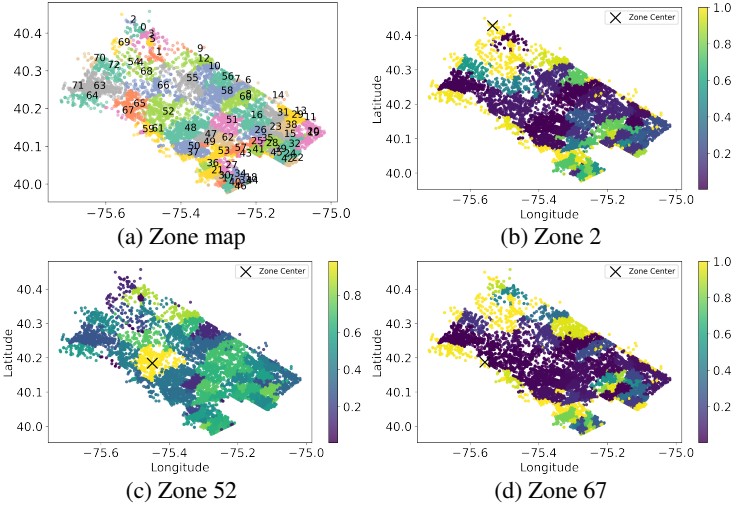

(a) Zone map  (b) Zone 2

(c) Zone 52  (d) Zone 67

Figure 5: The attention scores of zone 52, 2, and 67. The cross is the center of the zone.

score with them is high. This implies those events take significant impacts on updating the hidden state of the zone. However, there are also cases that the attention score with neighboring zones is small or close to zero, for example, the purple points surrounding Zone 2 (see Fig. 5b) and on right side of Zone 67 (see Fig. 5d). In such cases, the neighborhood events have little effect on the state update. This has shown the selection effect of the attention mechanism. On the other hand, the attentions score is in general smaller or closer to zero for distant zones. This might be attributed to the incorporation of the graph structure, because the distant vertexes are unlikely to be collected by the random walk, and hence their embeddings can be more dissimilar after training.

# 7 Conclusion

We have developed GINPP, a graph-informed neural point process that can avoid intractable integration in the likelihood, support multiple event types, and incorporate valuable graph knowledge into training and prediction. The experiments in ablation studies and seven real-world applications have shown the encouraging results.

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
