# OpenReview forum: "Graph-informed Neural Point Process With Monotonic Nets"
_ICLR.cc/2023/Conference — Submitted to ICLR 2023_

### Official Review · Reviewer_cb2m · 2022-10-13

**Confidence:** 3
**Correctness:** 3
**Technical Novelty And Significance:** 2
**Empirical Novelty And Significance:** 3
**Recommendation:** 5

**Clarity, Quality, Novelty And Reproducibility:**

Clarity:
The paper clearly describes its problem setting, and its proposed method.
Overall, the methods and the analysis are easy-to-follow.

Novelty:
It is new but the model in Eq.(6)(7) seems to be incremental.

Reproducibility:
It is not clear how to predict the event time and types in both Sec.4 (Algo1) and Sec. 6.2.

**Strength And Weaknesses:**

Strength
- The analysis of the training issue of the softplus function, looks interesting, and could be useful for later research.

Weaknesses
- The new model in Eq.(6)(7) seems to be incremental.
- It is not clear how to make time and event type prediction in Algo.1 of Sec.4.
The concrete equations for the time and type prediction, should be given !
- Why do not compare the predictive loglikelihood of the methods ?
- For the final experiments, why FullyNN (Omi 2019) is not included since you are circumventing its limitations?
- As I remember, the authors of transformer Hawkes processes also provide some solutions to incorporate graph among event sequences.
Can you compare your method to it?

**Summary Of The Paper:**

The paper considers a scenario in which there are M sequences of events, and each sequence consists of K event types.
The FullyNN based neural point process model (Omi 2019) demonstrates useful but needs to model each event type with a separate cumulative conditional intensity, which could be computationally demanding!
- To address this issue, this paper uses a separate function to explicitly model event type.
- The paper also analyzes the training difficulty of FullyNN induced by the SoftPlus function, and proposes a shift-scale variant of the softplus function, which demonstrates its effectiveness in the experiments.
- To leverage external graph information among sequences, they use an attention structure to modify how the recurrent neural nets' update its hidden states.

**Summary Of The Review:**

The paper mainly addresses a defect of FullyNN in modeling mixed-event types behind multiple related event sequences.
The novelty lies in modeling the event types with a separate neural net.
The analysis of the training issues of softplus function looks interesting, and could be very useful for latter research.
The paper uses an attention design to leverage external graph information among sequences, to better update RNN states, which further improve the model prediction accuracy.

Overall, the methods look interesting but appears to be incremental! The final experiments, only compared some known neural TPPs. I also suggest a few alternatives to make a fair comparison.

---

### Official Review · Reviewer_aMtU · 2022-10-23

**Confidence:** 4
**Correctness:** 3
**Technical Novelty And Significance:** 2
**Empirical Novelty And Significance:** 2
**Recommendation:** 6

**Clarity, Quality, Novelty And Reproducibility:**

The paper is well-written and easy to follow. The technical contributions are only marginally significant or novel.

**Strength And Weaknesses:**

Strength
1. The authors carry out empirical evaluation on a diverse set of real-world point processes with comparison to several state-of-art methods with superior performance.
2. The three improvements proposed in the new methods are well-justified with good motivation and dedicated analysis e.g. the scale shift-scale variant for the monotonic network output.
3. The authors carry out experiments on synthetic dataset to individually verify the effect of different components.

Weakness
1. The improvements to the previous method is a bit incremental. The marked process with conditional distribution is already widely used in point process modeling like Hawkes process. Similar to the incorporation of the neighbor information. The shift-scale variant output is also a standard trick for NN.
2. From the results in Table 1, it seems that the incorporation of graph information does not lead to better accuracy. Among two metrics on 2 datasets, GINPP is significantly better with only one setting (911Call RMSE). Moreover, it would be better for the authors to mark improvement with statistical significance.
3. It would be better if the authors could carry out single variable ablation experiments for the improvement (1) and (2). Currently, (1) is tested only partially on a monotonic net. It would be better to see the performance in the full NPP model. For (2), only the performance of the GINPP is presented. It would be interesting to see a comparison without the usage of conditional mark distribution.


**Summary Of The Paper:**

In this paper, the authors propose a new neural point process model with three improvements: (1) shift-scale variant for the output of monotonic net; (2) use marked process with conditional distribution; (3) incorporate graph information with random walk-based neighbor sampling. The authors compared the new method GINPP to several state-or-art methods on 6 real-world datasets.

**Summary Of The Review:**

The authors propose three practical improvement to NPP models with monotonic net. However, the technical contributions are only marginally significant or novel.

---

### Official Review · Reviewer_ou7C · 2022-10-24

**Confidence:** 4
**Correctness:** 2
**Technical Novelty And Significance:** 2
**Empirical Novelty And Significance:** 2
**Recommendation:** 3

**Clarity, Quality, Novelty And Reproducibility:**

The paper is written clearly and is easy to follow.

The proposed architectural improvements could be applied to other neural TPP models, but a more thorough ablation would be necessary to determine if they provide consistent improvements.

Some important aspects of the experimental setup (e.g., whether the data was normalized) are not mentioned and the code is not available, which could make reproducing the experimental results problematic.

**Strength And Weaknesses:**

Strengths:
- The paper is clearly written and is easy to follow.
- The proposed parametrization of the event type distribution given the time $p(s = k | t)$ provides an elegant and parameter-efficient way to model the intensity of marked TPPs. Conceptually, it offers clear advantages over models that model marks independently of the inter-event times.
- I really liked the inclusion of the "simple statistic" baseline in the experiments and wish more papers did the same.


Weaknesses:
1. Lack of ablations that convincingly demonstrate the effectiveness of each proposed modification to the model. It would be necessary to conduct experiments studying the effect of each individual modification. These would include comparisons to the original model from Omi et al.:
    - Compare the vanilla model from Omi et al. vs. model with the proposed shift-scale softplus activation on real-world dataset (showing both the convergence plots and the final RMSE/Accuracy results). The current experiment with recovering simple functions may not be representative of behavior on real-world event datasets.
    - Compare the proposed parametrization of $p(s_n = k | t)$ vs. the vanilla model from Omi et al. with conditionally independent marks $p(s_n = k)$ (as done, e.g., in [Du et al. 2016](https://www.kdd.org/kdd2016/papers/files/rpp1081-duA.pdf)).
    - The difference in performance between the model that incorporates the relational structure between event types (GINPP) and the model that ignores the graph (GINPP-1) in Table 1 is within the standard deviation of the scores in 3/4 cases. This brings into question whether the proposed attention mechanism improves the predictive performance.

The lack of these ablations makes it impossible to determine whether the overall good performance of the model in Table 1 is related to the proposed modifications, or if the original model by Omi et al. would achieve comparable results.

2. The novelty claim for the shift-scale softplus function ("we propose a shift-scale variant of the softplus function") could be adjusted. The scaled softplus function is quite standard (e.g., available in [PyTorch](https://pytorch.org/docs/stable/generated/torch.nn.Softplus.html)) and was even used in the context of TPPs in the original Neural Hawkes Process paper. It's also unclear whether the shift component of the softplus function is needed for the reported improvements due to the missing ablation. It could also be that simply normalizing the inter-event times before fitting the model would achieve similar improvements.


Minor comments:

3. Equations 5, 7: The first two terms in the second line should be $\exp(-\phi(...))$ instead of $\phi(...)$.
4. The statement "this method will increase the learning challenge for the monotonic net because K monotonic constraints have to be satisfied simultaneously" in Section 3 is not accurate. The monotonic constraints are satisfied by design of the neural network, so satisfying them simultaneously poses no challenge. Moreover, a recent work (https://dl.acm.org/doi/abs/10.1145/3511808.3557399) suggests that modeling a separate intensity for each mark can often improve the predictive performance.
5. In Section 6, clarify if we predict the type of the next event assuming the time is known (i.e., using $p(s_n = k | t = t_n)$) or without knowing the time (i.e., using $p(s_n = k)$).
6. Results for Poisson process (PP) in Table 1: The accuracy results should be identical for PP and SS, since the estimated intensity of the PP for the majority class should be higher than for other classes. Could it be that the PP model hasn't converged during training? Could other models be affected by the same issue?


**Summary Of The Paper:**

The paper introduces several modifications to the neural point process model from [Omi et al. 2019](https://arxiv.org/abs/1905.09690) that aim to improve the expresiveness and training speed of the model.
These improvements include:
- Passing the output of the monotonic neural network (modeling integrated intensity) through a shift-scale softplus activation function. This change leads to faster training of the model.
- Extending the model to marked TPPs with categorical marks by modeling the conditional distribution of the mark type given time $p(s = k | t)$ with a neural network.
- Incorporating the known relation structure between event types into the RNN update equation.


**Summary Of The Review:**

A more thorough empirical evaluation would be necessary to justify that all the proposed modifications to the model architecture indeed have a positive effect over the base model of Omi et al.

---

### Official Review · Reviewer_jeUB · 2022-10-24

**Confidence:** 5
**Clarity, Quality, Novelty And Reproducibility:** The paper is original and clearly pre…
**Correctness:** 4
**Technical Novelty And Significance:** 2
**Empirical Novelty And Significance:** 3
**Recommendation:** 5

**Strength And Weaknesses:**

In general, the paper has several strength. First, the paper is clearly written and the presentation is easy to follow. Second, the finding of the learning challenge in Omi et al. 2019 is significant. Modeling cumulative hazard function and calculating its derivative may result in unstable training process and poor predictive performance due to the softplus function, and the proposed shift-scale variant of the softplus function can indeed alleviate this problem.

While the extension of the Omi et al. 2019 is meaningful, it does not strike me as a fundamental modeling contribution to the point process literature, in that it does not offer a novel way to address a challenging modeling issue. The reason is two fold: 1. While being computationally efficient, decoupling lambda and mark term simplifies the model and dramatically limits the representative power; 2. The contribution of the variant of the softplus function is relatively incremental regarding the problem that this paper aims to address.

There are also a number of other previous studies attempting to address the same problem in different approaches without decoupling the mark and the conditional intensity, to name a few:
- https://proceedings.mlr.press/v157/hong21a.html
- https://openreview.net/forum?id=0rcbOaoBXbg.

**Summary Of The Paper:**

This paper investigated an important research problem in point process modeling and aimed to model multi-class event sequence in a network. As the paper pointed out, one of the recent work Omi et al. (2019) is known to be computationally efficient and flexible in representing complex triggering effects between events, however fails to model multi-class events and sometimes suffers from inefficient learning and easily falls into poor performance. The proposed method hinges on Omi et al. (2019) and extends the idea of modeling cumulative hazard function using monotonic net to support multiple event types. One of the most important assumption the proposed methods made is that the process has independent marks, i.e., the marks are conditionally independent of each other, so that the mark can be modeled separately. This simple modification enables efficient computation for both learning and inference. Another finding of this paper is that they investigate the learning challenge of Omi et al. 2019 and found that the bottleneck comes from the standard soft plus transformation at the output layer of the neural network, which causes large variance in the prediction. To alleviate this issue, the paper also proposes a shift-scale variant of the softplus function.

**Summary Of The Review:**

See above.

---

### Decision · Program_Chairs · 2023-01-20

**Decision:**

Reject

**Justification For Why Not Higher Score:**

Very limited significance of the contribution.

**Justification For Why Not Lower Score:**

N/A

**Metareview: Summary, Strengths And Weaknesses:**

The paper builds upon an existing temporal point process model by Omi et al. (2019) and addresses some of its shortcomings, particularly in terms of scalability. The reviewers agreed that the paper is well written and the methodology is sound. However, they all question the significance of the contribution is marginal/very limited and the experimental evaluation could somehow be improved. As a result, a majority of the reviewers were not in support of accepting the paper.